# Cultivating Leadership and Teamwork in Medical Students Through Rowing: A Phenomenological Study

**DOI:** 10.3390/bs14100962

**Published:** 2024-10-17

**Authors:** Hyo Jin Kwon, Su Jin Chae

**Affiliations:** Department of Medical Education, University of Ulsan College of Medicine, Seoul 05505, Republic of Korea; moksha@ulsan.ac.kr

**Keywords:** leadership, leadership training, rowing, teamwork, medical student, reflective journal

## Abstract

This study was conducted to allow us to understand the subjective experiences of medical students participating in rowing exercise classes at a medical school in South Korea and to derive implications for medical education. Accordingly, we analyzed their reflective journals, focusing on leadership and teamwork development. The study involved 40 second-year premedical students, and Colaizzi’s analysis was employed to understand and structure their experiences. The comprehensive analysis revealed 149 meaningful statements expressing students’ thoughts and experiences regarding the rowing exercise. From these statements, 13 meanings were synthesized, resulting in nine themes and four overarching categories, which provided a multilayered understanding of students’ experiences. The factors that enhanced teamwork included communication, trust, respect among team members, and a sense of responsibility. By contrast, the hindering factors were competitiveness, impatience, and avoidance of responsibility. Before the class, a mix of anticipation, excitement, and dissatisfaction regarding the rowing exercise course was observed. However, after the class, students realized that the role of the entire team, rather than individual ability, is crucial, and collaboration with peers is key—the concept of shared leadership. This study is significant in that it demonstrates rowing’s potential as a team sport to serve as an effective program for fostering collaboration and leadership within the medical school curriculum.

## 1. Introduction

With increasing specialization in medicine and the growing complexity of healthcare delivery systems, interest in integrating leadership and teamwork training into medical education has increased significantly [1]. Physicians must not only treat patients using their knowledge and skills but also act as leaders who communicate with and guide multidisciplinary healthcare teams, hospital departments, patients and their families, and fellow doctors. As a result, they are increasingly required to possess not only individual abilities but also competencies such as communication, teamwork, organizational skills, consideration, and service within a team context. These competencies are closely linked to leadership training [2]. Ford et al. [3] detail the intricate relationship between leadership and teamwork in the healthcare environment and emphasize that team efficiency is inextricably linked to patient safety outcomes. They also suggest that clinical training, including simulation exercises, is a representative example of team-based activities and is considered a form of leadership training in medical schools.

Some medical schools abroad have recognized the importance of leadership and have integrated leadership training into their curricula [4,5,6]. However, determining the specifics of the training to be implemented and how it should be carried out in medical schools still poses a challenge. This dilemma arises because the best approach to leadership education is to provide students with practical experiences rather than mere theoretical instruction. Several medical schools in South Korea also have curricula focused on developing leadership competencies, ultimately emphasizing communication and teamwork [7,8]. These leadership courses utilize various methods, including lectures, discussions, workshops, student presentations, written assignments, hospital experiences, and volunteer activities, all of which involve direct student participation. However, a review of the leadership education currently implemented in medical schools reveals that the number of programs related to sports activities is limited [9]. Research on general physical education (PE) programs at universities is also restricted to topics such as the development of a healthy self-identity [10], interpersonal relationships [11], and adjustment to university life [12].

This study provides a phenomenological exploration of the internal changes in teamwork and leadership among students at the University of Ulsan College of Medicine (UUCM) who participated in leadership training through team sports, specifically through rowing. Rowing emphasizes teamwork and requires a strong sense of commitment to the team [9]. Schools that participate in rowing competitions as intercollegiate events include Oxford and Cambridge, Harvard and Yale, and Waseda and Keio universities internationally, as well as Ulsan National Institute of Science and Technology, Korea Advanced Institute of Science and Technology, and Yonsei University in South Korea.

The educational goal of UUCM is to “cultivate future leaders who will lead the medical field”. To achieve this goal, a new curriculum was developed and implemented in 2022, which includes a leadership course utilizing rowing exercises. This study aims to discover the subjective experiences of second-year pre-medical students who participated in a rowing exercise class by analyzing the reflection journals they submitted as part of a leadership program. By understanding the students’ experiences in the rowing exercise class, the study seeks to derive implications for leadership education in medical schools.

## 2. Research Methodology

### 2.1. Study Participants

This study involved 40 second-year premedical students who participated in a leadership course from 23 May to 11 June 2022. After completing the rowing exercise, the students were asked to submit a reflective journal, approximately two A4 pages long, in which they freely expressed their thoughts on the rowing exercise. All 40 students submitted their reflective journals, which were subsequently analyzed.

### 2.2. Overview of the Rowing Exercise Course

UUCM is committed to its mission of “continuously striving for the healthy lives of humanity” and aims to cultivate medical leaders equipped with communication, ethics, and creativity skills. Since 2022, UUCM has implemented a new curriculum under the motto “Less Competitive, More Excellence” (LCME). This curriculum focuses on avoiding unnecessary and unproductive competition while fostering the essential competencies required of a good physician. An 11-week humanities and social sciences course is conducted for second-year premedical students, with the leadership course spanning two weeks. The content and schedule of the rowing exercise conducted as part of the leadership course are depicted in Figure 1.

The rowing exercise was conducted as part of the leadership course. The students were divided into Groups A and B, with 20 in each, and the exercise was conducted over one week for each group at the Misari Rowing Stadium. On the first day, the students attended a theoretical class on the importance of exercise and the history and techniques of rowing. On the second day, the students experienced a rowing machine workout at the Misari Rowing Stadium, along with warm-up exercises. The rowing machine was used indoors for aerobic and strength training to maintain the necessary sensations for rowing. On the third day, the students were divided into five teams to participate in a rowing practice session. Before boarding the boat, the students chose their roles freely (cox, seats 1, 2, 3, 4). After practicing rowing, they participated in team-based rowing races. On the final day of the course, the students submitted their reflective journals and participated in group presentations to share their thoughts and experiences.

### 2.3. Data Analysis

This study utilized Colaizzi’s phenomenological analysis procedure to understand and structure students’ experiences with the rowing exercise [13]. Colaizzi’s phenomenological analysis is a methodology employed by researchers to systematically analyze phenomenological data, aiming to understand the subjective experiences of research participants. The procedure involves several key steps: initially, meaningful statements related to the research phenomenon are identified and meanings are derived from these statements. Subsequently, these meanings are categorized, with the themes synthesized and integrated. Finally, the essential structure of the phenomenon is articulated as revealed through the research. In this study, Colaizzi’s method guided the data analysis, encompassing the phases of familiarization, meaning extraction, meaning clustering and thematization, and structure development [14].

Familiarization: Both researchers, experienced in qualitative research and phenomenological analysis, independently read and re-read all 40 reflective journals to capture the overall sentiment and identify key themes.

Meaning Extraction: Meaningful statements relating to the rowing experience and the development of leadership and teamwork were extracted from each journal. A detailed coding scheme was developed and consistently applied by both researchers.

Meaning Clustering and Thematization: Similar statements were grouped into clusters of meanings, which were then synthesized into broader themes. Any disagreements regarding the categorization of meanings and themes were resolved through discussion and consensus.

Structure Development: The themes were systematically integrated to construct an essential structure that reflects the comprehensive phenomenological experience of the rowing exercise, focusing on attitudes prior to the course, factors enhancing teamwork, hindering factors, and reflections following the course.

### 2.4. Ensuring Trustworthiness

To enhance the trustworthiness of the qualitative research findings, this study utilized prolonged engagement, triangulation, audit trail, and reflexivity.

Prolonged Engagement: To gain a deeper understanding of students’ experiences in the rowing-based leadership program and enhance data richness, the researchers participated in the two-week program, conducting an in-depth analysis of students’ reflective journals. This immersive approach facilitated a more precise understanding of students’ experiences and the transformative process they underwent.

Triangulation: To ensure trustworthiness, this study employed inter-rater reliability through triangulation. Two researchers with extensive experience in qualitative research independently analyzed the reflective journals. The researchers compared their individual analyses, identifying points of congruence and divergence. Disagreements were resolved through thorough discussion and consensus-building.

Audit Trail: To ensure transparency, a detailed research log was maintained, documenting the research questions, the literature review, phenomenological methodology, and the data analysis process (including data analysis records, coding, and interpretations). Furthermore, the coding data used in the study were made publicly available on the *FigShare* online repository to allow for scrutiny and potential use by other researchers.

Reflexivity: The researchers actively mitigated the potential influence of personal biases and values on the research findings. Regular research meetings facilitated the sharing of research processes and results, along with reciprocal feedback.

## 3. Results

Analysis of the experiences of medical students during rowing classes revealed 149 meaningful statements in the reflective journals. From these, 13 meanings were constructed, resulting in 9 themes and 4 categories. The categories, themes, and constructed meanings are summarized in Table 1.

### 3.1. Attitudes Before the Rowing Exercise Course

#### 3.1.1. Expectation and Excitement

Before participating, students experienced a mix of anticipation and concern. Although rowing was an unfamiliar sport, the fact that it was not something everyone could easily do excited them. They approached the course with a lighthearted attitude, which included looking forward to enjoying an activity outside the classroom, rather than simply learning theory. In addition, the announcement of team-based competitions gave them confidence, and they were eager to win.


*When we first saw rowing on the schedule, we were excited at the prospect of engaging in a fun activity with our peers. Rowing was a new sport that we had never tried before, and because it was something not typically experienced, it brought a great sense of anticipation and excitement.*



*We enjoyed breaking away from desk-based studying and engaging in outdoor activities because we had come to deeply appreciate the value of regular physical education classes, which were a part of our middle and high school years and which we missed after starting college. After becoming college students, we found that exercise had become an activity that required extra time and money.*


#### 3.1.2. Concerns and Complaints

Upon initial receipt of the course schedule, students questioned the inclusion of rowing exercises and, thus, highlighted the need to clarify the connection between this activity and medical education. Some students expressed dissatisfaction with having a physical exercise class in medical school, and others worried that their lack of physical strength might negatively affect their performance in the competition.


*We were initially filled with fear and worry when we first heard about the rowing exercise; it seemed like a sport that was difficult to see or experience, and as newcomers, we doubted whether it was something we could actually do.*



*We were quite surprised when we found out that rowing was part of the class. Our peers shared various opinions, such as it was added to fill the class schedule or the professor, who enjoys sports, included it to challenge the students.*


### 3.2. Factors Enhancing Teamwork

#### 3.2.1. Communication

Communication and providing feedback among team members are crucial in rowing. The students believed that constant verbal communication with their teammates was essential for synchronizing their movements during the rowing exercise.


*We realized the lead rower knows nothing if the rowers behind don’t communicate. If we had provided feedback to our teammates during the race, we could have gone faster than we did.*



*Our team finished in first place. Reflecting on how this was possible, the most decisive factor was our communication and teamwork.*


#### 3.2.2. Trust and Respect

In rowing, as the rowers face the opposite direction from the way the boat is moving, team members must pay close attention to each other’s movements and align themselves with their teammates.

Some team members need to take the lead, while others must demonstrate good followership to the leader. The students particularly emphasized the importance of harmony, synchronization, and the right tempo among team members.


*As we realized, the most important aspect of rowing is adjusting to each other. The person sitting at the front takes on the role of the leader, guiding the entire team. This person must maintain a steady pace and movement because the other team members rely on that to follow along. It’s important not to start rowing before the other members are ready or suddenly increase the pace just because you want to go faster. You must be considerate and ensure that others can keep up with you.*



*I believe that key elements of good followership include taking responsibility by following the leader’s decisions, providing appropriate feedback when necessary, and being tolerant of the leader’s mistakes while offering encouragement.*


#### 3.2.3. Responsibility

Achieving optimal speed in rowing requires each of the five team members to fulfill their specific roles diligently: the coxswain steers the boat, the fourth seat sets and maintains the rowing rhythm, the second and third seats follow the latter’s pace, and the first seat, which is positioned at the back, supports the entire team by observing and maintaining the rhythm. The students believed that the best way to make the boat go faster was for each person to take responsibility for their assigned role.


*I realized that rowing isn’t a sport where just one strong person can make a difference; it is a sport where everyone must watch each other and adjust to stay in sync. Moreover, in team activities like this, it’s easy to think, ‘It won’t matter if I just slack off,’ but rowing doesn’t allow for that kind of mindset.*


### 3.3. Factors Hindering Teamwork

#### 3.3.1. Competitiveness and Impatience

Unlike in practice, the students exhibited competitiveness and impatience during the race as they saw the other teams pulling ahead. Some students initially prioritized individual performance, thinking that it would be enough if they did well; however, this mindset contradicted the collaborative nature of the exercise. This personal view led to rowing without considering the pace of their teammates, ultimately causing them to fall behind the other teams.


*As it was a race, it was hard not to think about the speed of the other teams or our own team’s ranking. For this reason, the teammate in the fourth seat, who was responsible for regulating the overall speed of the boat, rowed too quickly out of impatience. The people behind couldn’t keep up with the fast pace that didn’t match the rhythm, which caused the oars to collide frequently and the team’s coordination to suffer. In the end, our team finished third out of the four teams.*



*I struggled to keep the rhythm and tried too hard to push, but this effort backfired. As our coordination continued to fall apart and we ended up in last place, I was really disappointed with the race’s outcome. Even if our ranking wasn’t great, I wouldn’t have been so disappointed if we had maintained our rhythm and pace throughout the race. We tried hard to do well, but the result was still disappointing.*


#### 3.3.2. Avoidance of Responsibility

The students found that avoiding or neglecting one’s duties can negatively impact the entire team in rowing, where each person is assigned a specific responsibility.


*There was a moment when I was rowing and got so exhausted that I stopped putting in effort. I was shocked to see how the boat almost stopped moving, and I felt bad for letting my team down, and realized I had been a burden to them.*


### 3.4. Reflections After the Course

#### Shared Leadership

After completing the rowing exercise, the students reflected on leadership within the team rather than as individuals. Accordingly, they realized the concept of leadership as a collective effort, where everyone works together toward a common goal, understanding the shared vision, and supporting one another along the way.


*Having teammates in front of and behind me, all working toward the same goal, was a great source of strength. It was an experience that made me realize how much the presence of my teammates could help me grow.*



*Through this rowing course, I realized collaboration can be more important than individual effort. I also learned how collaboration is developed and how other individuals supporting me can lead to effective teamwork.*


## 4. Discussion

The Association of American Medical Colleges [15] emphasizes the growing importance of leadership development in medical schools, stating that “leadership development has become more crucial than ever to anticipate, navigate, and address the complex challenges faced by medical schools and teaching hospitals today”. While numerous medical schools have developed and implemented leadership training programs focusing on communication and teamwork enhancement [16,17], the utilization of sports, particularly rowing, as a vehicle for leadership development represents a novel and underexplored approach in medical education.

The study analyzed the reflective journals written by pre-medical students who participated in rowing exercise classes conducted as part of a leadership course to understand their subjective experiences of the rowing classes. The analysis revealed the following results: first, before the rowing exercise course, students did not fully appreciate the value of incorporating physical activity into their medical education. Students viewed the course as a stress-free break from studying, that is, a time to relax and have fun. They felt a sense of anticipation and excitement about trying a sport they had never experienced. Meanwhile, they were also concerned about not performing well and whether the exercise might be too physically demanding. In addition, some dissatisfaction was expressed about why such a sport was included in the medical school curriculum. Medical students lead competitive lives even before entering medical school, particularly in preparation for their admission. After entering medical school, students experience significant amounts of coursework, pressure to maintain high grades, feelings of relative failure, and academic stress due to intense competition [18]. They also often suffer from issues such as sleep deprivation and chronic fatigue [19]. Previous research [20] documents that medical students’ perfectionist tendencies lead them to set excessively high standards for themselves and to strive for perfection. Furthermore, they tend to be highly self-critical when they fail to meet these standards.

According to previous studies [21], sports education can contribute to students’ health as well as their interpersonal relationships, personal psychosocial skills, and professionalism. Therefore, there is a need for educational programs that promote mental and physical health for medical students who are exposed to competition and experience high levels of stress. However, to date, there has been a lack of basic research in South Korea on how much physical activity education medical students are engaging in at university and what types of exercise classes are included in the official curriculum. It is necessary to provide reasons as to why educational programs focused on physical activity are needed for medical students, as well as to design effective educational plans for the implementation of these programs. Additionally, follow-up studies should be conducted on the effectiveness of educational programs utilizing sports in medical education.

Second, through their participation in the rowing program, students gained a deeper understanding of the factors that both foster and impede effective teamwork. In their reflective journals, the factors that enhanced teamwork included communication, trust and consideration, and a sense of responsibility, while the hindering factors were competitiveness and impatience, arrogance about individual ability, and avoidance of responsibility. Rowing is characterized by the fact that the boat can only move forward through cooperation and harmony; no matter how skilled someone is, they cannot move the boat on their own. The rowing course allowed even students accustomed to individualism to reflect on how working together could lead to greater efficiency and the ability to go further. Similarly, our findings align with those of Yang [9], who also investigated rowing exercises at a medical school. Students developed a sense of community as they synchronized their movements, thereby fostering a mindset of consideration and respect for one another. In addition, in team-based general PE courses at universities, students developed confidence in their ability to complete tasks with their peers, which led to improved academic performance [22].

Our findings strongly suggest that integrating team sports, such as rowing, into medical education curricula can significantly enhance students’ collaborative skills and their capacity to achieve collective goals, potentially improving their performance in multidisciplinary healthcare teams. Because non-cognitive competencies such as communication, trust and respect, and responsibility are difficult to develop in a short period, there is a need for a curriculum that can systematically cultivate non-cognitive abilities that medical students should possess through various activities, including team sports.

Third, through their rowing experience, students developed a new perspective on leadership. They learned that effective leadership emphasizes collaboration and teamwork, with the leader’s role being to facilitate collective effort and progress rather than to rely solely on individual skills. The leader’s role is even more critical in a rowing race because multiple individuals must work together to move the boat. The situation is similar for physicians, who need to collaborate with various professionals. Wiseman et al. [2] identified critical leadership competencies essential for residency training, encompassing responsibility, teamwork, moral integrity, and empathy, which align with the skills fostered through the rowing exercise. Additionally, Stillman [23], who studied the relationship between leadership and team performance, reported that leadership is a critical element for the success of an organization and is important for medical staff collaborating with various professions, positively impacting team performance.

Leadership is often defined as the skill of exerting influence over others to motivate them toward achieving a common goal [24]. To date, most studies on leadership in medical schools have concentrated on self-leadership [25], which focuses on motivating oneself and taking responsibility through autonomy and self-control, ultimately exerting influence over one’s actions and decisions. However, shared leadership emphasizes the roles of the entire team rather than a single leader. Shared leadership is one of the leadership approaches adopted by many organizations to address the complexities of modern organizations, referring to a team phenomenon where the role and influence of the leader are distributed among team members [26].

The study found that students experienced the importance of mutual influence among members for the group’s goals rather than individual achievement through the rowing exercise class. In other words, students showed a shift in the concept of leadership, valuing the ‘team’ over ‘self’ through the rowing exercise. In rowing, each team member is assigned specific roles and responsibilities. Students found that while individual abilities are important for maximizing team effectiveness, trust among team members who can achieve common goals and tasks together is crucial. The rowing class provided students with an opportunity to reflect on how the team can grow and develop through mutual feedback from team members. In medical education, shared leadership has traditionally been studied in clinical settings such as emergency situations or simulation training [27]. This is because simulation training is similar to a sport in which multiple team members work together towards the common goal of patient safety. However, the results of this study indicate that leadership skills in medical students can be developed not only through simulation training limited to clinical practice but also through team sports. Medical schools need to develop and implement curricula that incorporate a variety of activities, including team sports, to provide more meaningful leadership education.

The significance of our study lies in its demonstration of the potential of rowing as a team sport to serve as an effective program for fostering teamwork and leadership within the medical school curriculum. As this study only analyzed the reflective journals of second-year premedical students from a single university who participated in the rowing exercise course, it has limitations in terms of generalizing the results. Future research should analyze students’ experiences using various methods, including not only the analysis of reflective journals but also student interviews and surveys. Moreover, the experiences of students at other medical schools that have incorporated various team sports into their curricula need to be investigated, and the findings used to inform the development of medical education programs.

## 5. Conclusions

This study phenomenologically analyzed the subjective experiences of medical students participating in rowing to examine the effectiveness of using team sports as a leadership development program in medical education. The findings revealed that rowing had a positive impact on medical students’ teamwork, communication skills, trust-building, and the development of shared leadership. Notably, the rowing process fostered an understanding of the importance of communication and cooperation, leading to improved relationships among team members. This emphasizes the significance of shared leadership, where collective efforts are valued over individual capabilities in achieving common goals. This study suggests the potential for incorporating team sports into medical education for leadership development. Rowing can be an effective program for enhancing teamwork and shared leadership skills among medical students, indicating its potential for nurturing future leaders in the medical field.

## Figures and Tables

**Figure 1 behavsci-14-00962-f001:**
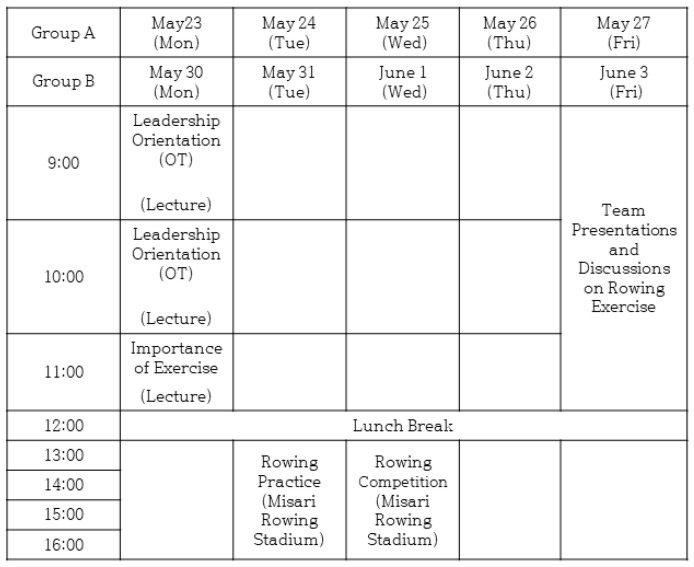
Rowing exercise schedule.

**Table 1 behavsci-14-00962-t001:** The essential structure of the experience of rowing exercise classes.

Categories	Themes	Constructed Meanings
Attitudes before the Rowing Exercise Course	Expectation and Excitement	Enjoyment and excitement
		Confidence in doing well
	Concerns and Complaints	Worry about being physically weak
		Uncertainty about the purpose of this course
Factors Enhancing Teamwork	Communication	Importance of communication among team members
	Trust and Respect	Need to synchronize and align with each other
		Necessity of being a good follower
	Responsibility	Everyone must do their best in their respective roles
Factors Hindering Teamwork	Competitiveness and Impatience	Desire to win
	Arrogance about Individual Ability	Prioritizing personal ability over the team’s needs
	Avoidance of Responsibility	“It’s OK if I just slack off” attitude
Reflections after the Rowing Exercise Course	Shared Leadership	Viewing teammates as collaborators, not competitors
		Joy of working together

## Data Availability

The data presented in this study are openly available on FigShare at the following URL: https://figshare.com/s/190bc6edba9b4c7b59c5, accessed on 29 August 2024.

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
