# Peer review of "Cultivating Leadership and Teamwork in Medical Students Through Rowing: A Phenomenological Study"

_behavsci, 2024, doi:10.3390/bs14100962_

Round 1
Reviewer 1 Report
Comments and Suggestions for Authors
1. Abstract = Check journal format and make it structured.
2. Introduction = Objectives have not been explicitly mentioned. Instead of Objectives, authors have mentioned research questions and that makes it not clear
3. Methodology = Major corrections desired
Mention what all precautions / steps were taken by the researchers to maintain trustworthiness (Credibility, Dependability, Transferability, Confirmability). Do not forget to include about reflexivity
Even no details have been mentioned about Transcription - Who did? How? etc.
4. Discussion = Authors must aim to "discuss" the results (why similar or contrasting results were obtained) and not just cite
5. There is a scope for improving the writing style.
Comments on the Quality of English LanguageImprove the english... Ending research question with "Question Mark"
Author Response
Thank you for the opportunity to revise and resubmit our manuscript, "Cultivating Leadership and Teamwork in Medical Students through Rowing: A Phenomenological Study". We appreciate the insightful and constructive comments provided by the reviewers, which have significantly helped us improve the quality of our work.
We have provided responses to the reviewer's comments on a point-by-point basis. The responses are included in the attached Word file. Please review the contents of the file.
We have carefully considered all the feedback and have made the necessary revisions.
We believe these revisions have strengthened our manuscript, and we hope it now meets the expectations.
Thank you, again.

Reviewer 2 Report
Comments and Suggestions for Authors
This is a study with an original topic, which will contribute to the under-researched field. My questions are related to the methodology:
1. This reads like a generic/basic qualitative study or thematic analysis of students' reflection.
2. If this is a phenomenology, what is the phenomenon/ the "essence"/ "essential structure" you want to explore in-depth. Is collected data suitable for this kind of analysis?
3. The Colaizzi’s phenomenological analysis procedure was mentioned, but not explained. How did you apply his approach when analyzing the reflections?
4. Ln 106-107 "Meaningful statements related to the research phenomenon were extracted, and meanings were formulated from the key statements." Ln 108-109 "Throughout this process, the original text was repeatedly reviewed, formulated meanings ..." Ln 119-120 "The analysis revealed 149 meaningful statements in the reflective journals. From these, 13 meanings were constructed..." If this is a pheno, an explanation of "meaning" is needed. Who formulated the meaning? The students, the researchers? The experiences were meaningful to whom? Is this meaning the essence of the phenomenon?
Thank you.
Comments on the Quality of English LanguageI assume all the papers are run by a copy editor.
Author Response

(The authors gave the same response as above.)

Round 2
Reviewer 1 Report
Comments and Suggestions for Authors My comments pertaining to Trustworthiness and Discussion have not been addressed satisfatorilyFor instance, credibility can be enhanced through Prolonged engagement, Triangulation, Member Checking, Reflexivity, Audit trail, etc...
Similarly, all 4 trustworthiness parameters should be justified
Discussion is again mere citing of results from your study and other study. No potential explanation cited for obtaining similar or contrasting results
Author Response
Comments 1: My comments pertaining to Trustworthiness and Discussion have not been addressed satisfactorily. For instance, credibility can be enhanced through Prolonged engagement, Triangulation, Member Checking, Reflexivity, Audit trail, etc...Similarly, all 4 trustworthiness parameters should be justified
Response 1: Thank you for your review comments.
2.3 Revision of the data analysis content.
In this study, Colaizzi's method guided the data analysis, encompassing the phases of familiarization, meaning extraction, meaning clustering and thematization, and structure development [14].
Familiarization: Both researchers, experienced in qualitative research and phenomenological analysis, independently read and re-read all 40 reflective journals to capture the overall sentiment and identify key themes.
Meaning Extraction: Meaningful statements relating to the rowing experience and the development of leadership and teamwork were extracted from each journal. A detailed coding scheme was developed and consistently applied by both researchers.
Meaning Clustering and Thematization: Similar statements were grouped into clusters of meanings, which were then synthesized into broader themes. Any disagreements regarding the categorization of meanings and themes were resolved through discussion and consensus.
Structure Development: The themes were systematically integrated to construct an essential structure that reflects the comprehensive phenomenological experience of the rowing exercise, focusing on attitudes prior to the course, factors enhancing teamwork, hindering factors, and reflections following the course.
2.4 Addition of the area for ensuring trustworthiness.
To enhance the trustworthiness of the qualitative research findings, this study utilized prolonged engagement, triangulation, audit trail, and reflexivity.
Prolonged Engagement: To gain a deeper understanding of students' experiences in the rowing-based leadership program and enhance data richness, the researchers participated in the two-week program, conducting an in-depth analysis of students' reflective journals. This immersive approach facilitated a more precise understanding of students' experiences and the transformative process they underwent.
Triangulation: To ensure trustworthiness, this study employed inter-rater reliability through triangulation. Two researchers with extensive experience in qualitative research independently analyzed the reflective journals. The researchers compared their individual analyses, identifying points of congruence and divergence. Disagreements were resolved through thorough discussion and consensus-building.
Audit Trail: To ensure transparency, a detailed research log was maintained, documenting the research questions, literature review, phenomenological methodology, and the data analysis process (including data analysis records, coding, and interpretations). Furthermore, the coding data used in the study were made publicly available on the Figshare online repository to allow for scrutiny and potential use by other researchers.
Reflexivity: The researchers actively mitigated the potential influence of personal biases and values on the research findings. Regular research meetings facilitated the sharing of research processes and results, along with reciprocal feedback.
Comments 2: Discussion is again mere citing of results from your study and other study. No potential explanation cited for obtaining similar or contrasting results
Response 2: The authors appreciate the reviewers' comments. Instead of simply listing research findings, the authors have revisited and summarized the students' internal transformations resulting from their rowing experience, interpreting these findings in light of prior research and offering recommendations for improvement in South Korean medical education. Significant revisions are highlighted in red in the manuscript. These revisions detail the changes observed in the student.
First, before the rowing exercise course, students did not fully appreciate the value of incorporating physical activity into their medical education.
Second, through their participation in the rowing program, students gained a deeper understanding of the factors that both foster and impede effective teamwork
Third, through their rowing experience, students developed a new perspective on leadership. They learned that effective leadership emphasizes collaboration and teamwork, with the leader's role being to facilitate collective effort and progress rather than to rely solely on individual skills.

Reviewer 2 Report
Comments and Suggestions for Authors
Thank you for the revisions!
Author Response
Thank you sincerely for providing the review results. Thanks to your comments, we have revised our paper into a deep and high-quality piece. Thank you once again

Round 3
Reviewer 1 Report
Comments and Suggestions for Authors
Suggested corrections done. Satisfied with the modifications made.